# Optimizing Learning Rate Schedules
# for Iterative Pruning of Deep Neural Networks

**Shiyu Liu**                                                                   *shiyu_liu@u.nus.edu*
*Department of Electrical and Computer Engineering*
*College of Design and Engineering*
*National University of Singapore*

**Rohan Ghosh**                                                                 *rghosh92@gmail.com*
*Department of Electrical and Computer Engineering*
*College of Design and Engineering*
*National University of Singapore*

**John Chong Min Tan**                                                          *johntancm@u.nus.edu.sg*
*Department of Electrical and Computer Engineering*
*College of Design and Engineering*
*National University of Singapore*

**Mehul Motani**                                                                *motani@nus.edu.sg*
*Department of Electrical and Computer Engineering*
*College of Design and Engineering*
*N.1 Institute for Health*
*Institute of Data Science*
*Institute for Digital Medicine (WisDM)*
*National University of Singapore*

**Reviewed on OpenReview:** *https://openreview.net/forum?id=nGW2Hotpq3*

## Abstract

The importance of learning rate (LR) schedules on network pruning has been observed in a few recent works. As an example, *Frankle and Carbin (2019)* highlighted that winning tickets (i.e., accuracy preserving subnetworks) can not be found without applying a LR warmup schedule. *Renda, Frankle and Carbin (2020)* also demonstrated that rewinding the LR to its initial state at the end of each pruning cycle can improve pruning performance. In this paper, we go one step further by first providing a theoretical justification for the surprising effect of LR schedules. Next, we propose a LR schedule for network pruning called SILO, which stands for S-shaped Improved Learning rate Optimization. The advantages of SILO over existing LR schedules are two-fold: (i) SILO has a strong theoretical motivation and dynamically adjusts the LR during pruning to improve generalization. Specifically, SILO increases the LR upper bound (`max_lr`) in an S-shape. This leads to an improvement of 2% - 4% in extensive experiments with various types of networks (e.g., Vision Transformers, ResNet) on popular datasets such as ImageNet, CIFAR-10/100. (ii) In addition to the strong theoretical motivation, SILO is empirically optimal in the sense of matching an Oracle, which exhaustively searches for the optimal value of `max_lr` via grid search. We find that SILO is able to precisely adjust the value of `max_lr` to be within the Oracle optimized interval, resulting in performance competitive with the Oracle with significantly lower complexity.

# 1 Introduction

Network pruning is the process of simplifying neural networks by pruning weights, filters or neurons. (LeCun et al., 1990; Han et al., 2015b). Several state-of-the-art pruning methods (Renda et al., 2019; Frankle & Carbin, 2019) have demonstrated that a significant quantity of parameters can be removed without sacrificing accuracy. This greatly reduces the resource demand of neural networks, such as storage requirements and energy consumption (He et al., 2020; Wang et al., 2021; Good & et al, 2022; Vysogorets & Kempe, 2023).

The inspiring performance of pruning methods hinges on a key factor - Learning Rate (LR). Specifically, Frankle & Carbin (2019) proposed the Lottery Ticket Hypothesis and demonstrated that the winning tickets (i.e., the pruned network that can train in isolation to full accuracy) can not be found without applying a LR warmup schedule. In a follow-up work, Renda et al. (2019) proposed LR rewinding which rewinds the LR schedule to its initial state during iterative pruning and demonstrated that it can outperform standard fine-tuning. In summary, the results in both works suggest that, besides the pruning metric, LR also plays an important role in network pruning and could be another key to improving the pruning performance.

In this paper, we take existing studies one step further and aim to optimize the choice of LR for iterative network pruning. We explore a new perspective on adapting the LR schedule to improve the iterative pruning performance. In the following, we summarize the workflow of our study together with our contributions.

1. **Motivation and Theoretical Study.** In Section 3.1, we explore the optimal choice of LR during network pruning and find that the distribution of weight gradients tends to become narrower during pruning, suggesting that a larger value of LR should be used to retrain the pruned network. This finding is further verified by our theoretical development in Section 3.2. More importantly, our theoretical results suggest that the optimal increasing trajectory of LR should follow an S-shape.

2. **Proposed SILO.** In Section 4, we propose a novel LR schedule for network pruning called SILO, which stands for S-Shaped Improved Learning rate Optimization. Motivated by our theoretical development, SILO precisely adjusts the LR by increasing the LR upper bound (`max_lr`) in an S-shape. We highlight that SILO is method agnostic and works well with numerous pruning methods.

3. **Experiments.** In Section 5.2, we compare SILO to four LR schedule benchmarks via both classical and state-of-the-art (SOTA) pruning methods. We observe that SILO outperforms LR schedule benchmarks, leading to an improvement of 2% - 4% in extensive experiments with various networks (e.g., Vision Transformer (Dosovitskiy et al., 2020), ResNet (He et al., 2016), VGG (Simonyan & Zisserman, 2014), DenseNet-40 (Huang et al., 2017) and MobileNetV2 (Sandler et al., 2018)) on large-scale datasets such as ImageNet (Deng et al., 2009) and popular datasets such as CIFAR-10, CIFAR-100 (Krizhevsky et al., 2009).

4. **Comparison to Oracle.** In Section 5.4, we examine the optimality of SILO by comparing it to an Oracle which exhaustively searches for the optimal value of `max_lr` via grid search. We find that SILO is able to precisely adjust `max_lr` to be within the Oracle's optimized `max_lr` interval at each pruning cycle, resulting in performance competitive with the Oracle, but with significantly lower complexity.

# 2 Background

In Section 2.1, we first review prior works on network pruning. Next, in Section 2.2, we highlight the important role of LR in network pruning and position our work in the context of network pruning.

## 2.1 Prior Works on Network Pruning

Network pruning is an established idea dating back to 1990 (LeCun et al., 1990). The motivation is that networks tend to be overparameterized and redundant weights can be removed with a negligible loss in accuracy (Arora et al., 2018; Allen-Zhu et al., 2019; Denil et al., 2013). Given a trained network, one **pruning cycle** consists of three steps: (i) Prune the network according to certain heuristics; (ii) Freeze pruned parameters as zero. (iii) Retrain the pruned network to recover the accuracy. Repeating the pruning cycle multiple times until the target sparsity or accuracy is met is known as **iterative pruning**. Doing so often results in better performance than **one-shot pruning** (i.e., perform only one pruning cycle) (Han

et al., 2015b; Frankle & Carbin, 2019; Li et al., 2017). There are two types of network pruning - unstructured pruning and structured pruning - which will be discussed in detail below.

**Unstructured Pruning** removes individual weights according to certain heuristics, such as magnitude (Han et al., 2015b; Frankle & Carbin, 2019) or gradient (Hassibi & Stork, 1993; LeCun et al., 1990; Lee et al., 2019; Xiao et al., 2019; Theis et al., 2018). Examples are (LeCun et al., 1990), which performed pruning based on the Hessian Matrix, and (Theis et al., 2018), which used Fisher information to approximate the Hessian Matrix. Similarly, Han et al. (2015b) removed weights with the smallest magnitude, and this approach was further incorporated with the three-stage iterative pruning pipeline in (Han et al., 2015a).

**Structured Pruning** involves pruning weights in groups, neurons, channels or filters (Yang et al., 2019; Molchanov et al., 2017; 2019; Luo et al., 2017; Yu et al., 2018; Tan & Motani, 2020; Wang et al., 2020b; Lin et al., 2020; Zhang & Freris, 2023). Examples are (Hu et al., 2016), which removed neurons with high average zero output ratio, and (Li et al., 2017), which pruned neurons with the lowest absolute summation values of incoming weights. More recently, Yu et al. (2018) proposed the neuron importance score propagation algorithm to evaluate the importance of network structures. Molchanov et al. (2019) used Taylor expansions to approximate a filter's contribution to the final loss and Wang et al. (2020a) optimized the neural network architecture, pruning policy, and quantization policy together in a joint manner.

**Other Works.** In addition to works mentioned above, several other works also share some deeper insights in network pruning (Liu et al., 2019b; Wang et al., 2020c; Li & et al, 2022; Wang & et al, 2022). For example, Liu et al. (2019a) demonstrated that training-from-scratch on the right sparse architecture yields better results than pruning from pre-trained models. Similarly, Wang et al. (2020c) suggested that the fully-trained network could reduce the search space for the pruned structure. More recently, Luo & Wu (2020) addressed the issue of pruning residual connections with limited data and Ye et al. (2020) theoretically proved the existence of small subnetworks with lower loss than the unpruned network. You & et al (2022) motivated the use of the affine spline formulation of networks to analyze recent pruning techniques. Liu et al. (2022) applied the network pruning technique in graph networks and approximated the subgraph edit distance. One milestone paper (Frankle & Carbin, 2019) pointed out that re-initializing with the original parameters (known as weight rewinding) plays an important role in pruning and helps to further prune the network with negligible loss in accuracy. Some follow-on works (Zhou et al., 2019; Renda et al., 2019; Malach et al., 2020) investigated this phenomenon more precisely and applied this method in other fields (e.g., transfer learning (Mehta, 2019) and natural language processing (Yu et al., 2020)).

## 2.2 The Important Role of Learning Rate

Several recent works (Renda et al., 2019; Frankle & Carbin, 2019) have noticed the important role of LR in network pruning. For example, Frankle & Carbin (2019) demonstrated that training VGG-19 with a LR warmup schedule (i.e., increase LR to 1e-1 and decrease it to 1e-3) and a constant LR of 1e-2 results in comparable accuracy for the unpruned network. However, as the network is iteratively pruned, the LR warmup schedule leads to a higher accuracy (see Fig.7 in (Frankle & Carbin, 2019)). In a follow-up work, Renda et al. (2019) further investigated this phenomenon and proposed a retraining technique called LR rewinding which can always outperform the standard retraining technique called fine-tuning (Han et al., 2015b). The difference is that fine-tuning trains the unpruned network with a LR warmup schedule, and retrains the pruned network with a constant LR (i.e., the final LR of the schedule) in subsequent pruning cycles (Liu et al., 2019b). LR rewinding retrains the pruned network by rewinding the LR warmup schedule to its initial state, namely that LR rewinding uses the same schedule for every pruning cycle. As an example, they demonstrated that retraining the pruned ResNet-50 using LR rewinding yields higher accuracy than fine-tuning (see Figs.1 & 2 in (Renda et al., 2019)). In (Liu & et al, 2021), the authors also suggest that when pruning happens during the training phase with a large LR, models can easily recover from pruning than using a smaller LR. Overall, The results in these works suggest that, besides the pruning metric, LR also plays an important role in network pruning and could be another key to improving network pruning.

**Our work.** In this paper, we explore a new perspective on adapting the LR schedule to improve the iterative pruning performance of ReLU-based networks. The proposed LR schedule is method agnostic and can work well with numerous pruning methods. We mainly focus on iterative pruning of ReLU-based networks for two

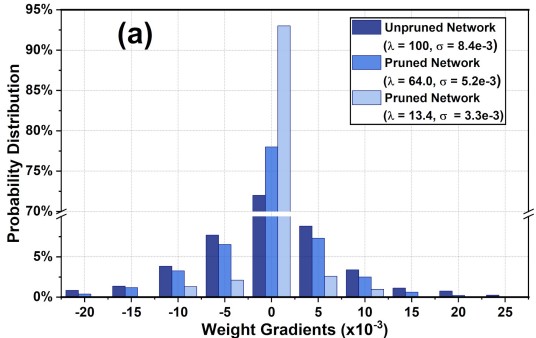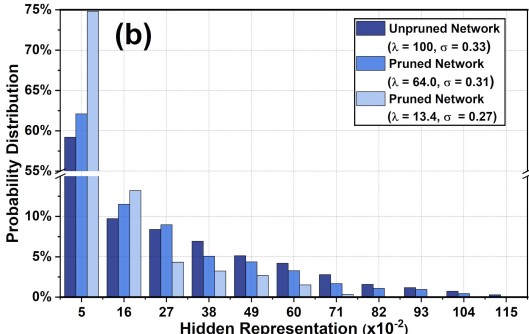

Figure 1: (a) The distribution of all weight gradients when iteratively pruning a fully connected ReLU network using global magnitude, where $\lambda$ is the percent of weights remaining and $\sigma$ is the standard deviation of the distribution. (b) The corresponding distribution of hidden representations.

reasons: (i) Iterative pruning tends to provide better pruning performance than one-shot pruning as reported in the literature (Frankle & Carbin, 2019; Renda et al., 2019). (ii) ReLU has been widely used in many classical neural networks (e.g., ResNet, VGG, DenseNet) which have achieved outstanding performance in various tasks (e.g., image classification, object detection) (He et al., 2016; Simonyan & Zisserman, 2014).

## 3 A New Insight on Network Pruning

In Section 3.1, we first provide a new insight in network pruning using experiments. Next, in Section 3.2, we provide a theoretical justification for our observed new insight and present some relevant theoretical results.

### 3.1 Weight Gradients during Iterative Pruning

**(1) Experiment Setup.** To exclude the influence of other factors, we start from a simple fully connected ReLU-based network with three hidden layers of 256 neurons each (results of other popular networks are summarized later). We train the network using the training dataset of CIFAR-10 via SGD (Ruder, 2016) (momentum = 0.9 and a weight decay of 1e-4) with a batch size of 128 for 500 epochs. All hyperparameters are tuned for performance via grid search (e.g., LR from 1e-4 to 1e-2). We apply the global magnitude (Han et al., 2015b) (i.e., remove weights with the smallest magnitude anywhere in the network) with a pruning rate of 0.2 (i.e., prune 20% of the remaining parameters) to iteratively prune the network for 10 pruning cycles and plot the distribution of all weight gradients when the network converges in Fig. 1(a), where $\lambda$ is the percent of weights remaining. In Fig. 1(a), there are 10 visible bins estimated by the Sturges' Rule (Scott, 2009) and each bin consists of three values (i.e., the probability distribution of networks with different $\lambda$). The edge values range from -0.022 to 0.027 with a bin width of 0.004.

**(2) Experiment Results.** In Fig. 1(a), we observe that the distribution of weight gradients tends to become narrower, i.e., the standard deviation of weight gradients $\sigma$ reduces from 8.4e-3 to 3.3e-3 when the network is iteratively pruned to $\lambda = 13.4$. As an example, the unpruned network ($\lambda = 100$) has more than 7% of weight gradients with values greater than 0.008 (rightmost 4 bars) or less than -0.012 (leftmost 2 bars), while the pruned network ($\lambda = 13.4$) has less than 1% of weight gradients falling into those regions. It suggests that the magnitude of weight gradients tends to decrease as the network is iteratively pruned.

**(3) New Insight.** During the backpropagation, the weight update of $w_i$ is $w_i \leftarrow w_i + \alpha \frac{\partial \mathcal{L}}{\partial w_i}$, where $\alpha$ is the LR and $\mathcal{L}$ is the loss function. Assume that $\alpha$ is well-tuned to ensure the weight update (i.e., $\alpha \frac{\partial \mathcal{L}}{\partial w_i}$) is sufficiently large to prevent the network from getting stuck in local optimal points (Bengio, 2012; Goodfellow et al., 2016). As shown in Fig. 1(a), the magnitude of the weight gradient (i.e., $\frac{\partial \mathcal{L}}{\partial w_i}$) tends to decrease as the network is iteratively pruned. To preserve the same weight updating size and effect as before, a gradually larger value of LR (i.e., $\alpha$) should be used to retrain the pruned network during iterative pruning.

**(4) Result Analysis.** We now provide an explanation for the change in the distribution of weight gradients. First, we assume each $x_i w_i$ (i.e., $x_i \in \mathbb{R}$ is the neuron input and $w_i \in \mathbb{R}$ is the associated weight) is an i.i.d. random variable. Then, the variance of the neuron's pre-activation output ($\sum_{i=1}^{n} x_i w_i$, $n$ is the number of inputs) will be $\sum_{i=1}^{n} \text{Var}(x_i w_i)$. Pruning the network is equivalent to reducing the number of inputs from $n$ to $n - k$. This results in a smaller variance of $\sum_{i=1}^{n-k} \text{Var}(x_i w_i)$, leading to a smaller standard deviation. Hence, the distribution of the pre-activation output after pruning is narrower. Since ReLU returns its raw input if the input is non-negative, the distribution of hidden representations (output of hidden layers) becomes narrower as well. This can be verified from Fig. 1(b), where we plot the distribution of hidden representations from the previous experiment. The key is that the weight gradient $\frac{\partial \mathcal{L}}{\partial w_i}$ is proportional to the hidden representation $x_i$ that associates with $w_i$ (i.e., $\frac{\partial \mathcal{L}}{\partial w_i} \propto x_i$). As the network is iteratively pruned, the distribution of hidden representations becomes narrower, leading to a narrower distribution of weight gradients. As a result, a larger LR should be used to retrain the pruned network.

**(5) More Generalized Results.** (i) Effect of Batch Normalization (BN) (Ioffe & Szegedy, 2015): BN is a popular technique to reformat the distribution of hidden representations, so as to address the issue of internal covariate shift. We note that similar performance trends can be observed after applying BN as well (see Fig. 4 in the Appendix). (ii) Popular CNN Networks and Pruning Methods: In addition to the global magnitude used before, two unstructured pruning methods (i.e., layer magnitude, global gradient) suggested by (Blalock et al., 2020) and one structured pruning method (L1 norm pruning) (Li et al., 2017) are examined as well. Those methods are used to iteratively prune AlexNet (Krizhevsky & Hinton, 2010), ResNet-20 and VGG-19 using CIFAR-10. The results using these popular neural networks largely mirror those in Figs. 1(a) & (b) as well. We refer the interested reader to Figs. 5 - 7 in the Appendix.

## 3.2 Theoretical Study and Motivation

In this subsection, we theoretically investigate how network pruning can influence the value of the desired LR. The proofs of the results given here are provided in the Appendix. First, we present some definitions.

**Definition 1. Average Activation Norm ($E_{AA}$):** Given a network with fixed weights, input $X$ from a distribution $P$, and a layer $H = \{h_1(X), ..., h_N(X)\}$ with $N$ nodes where $h_i(X)$ represents the function at the $i^{th}$ node. Then $E_{AA}(H) = \mathbb{E}_X[\frac{1}{N} \sum_i h_i(X)^2]$. This quantity reflects the average strength of the layer's activations.

**Definition 2. Average Gradient Norm ($E_{WG}$):** Let $W = [w_1, ..., w_k]$ and $W' = [w_1', ..., w_k']$ represent the flattened weight vector before and after one epoch of training (via backpropagation). Then $E_{WG}$ is the average change in weight magnitude in a single training epoch (for the active unpruned weights), i.e., $E_{WG}(W, W') = \mathbb{E}_i[(w_i - w_i')^2]$. $E_{WG}$ quantifies how much the weights change after one epoch of training.

We now demonstrate the impact of network pruning on the *average activation norm* of hidden layers.

**Theorem 1.** Consider a ReLU activated neural network represented as $X \xrightarrow{W_1} H \xrightarrow{W_2} Y$, where $X \in \mathbb{R}^d$ is the input, $H = \{H_1(X), H_2(X), ..., H_N(X)\}$ is of infinite width ($N = \infty$), and $Y$ is the network output. $W_1$ and $W_2$ represent network parameters (weights, biases). Furthermore, let $X \sim \mathcal{N}(0, \sigma_x^2 I)$ and $W_1 \sim \mathcal{N}(0, \sigma_w^2 I)$, where $I$ is the identity matrix and $\sigma_x, \sigma_w$ are scalars. Now, let us consider an iterative pruning method, where in each iteration a fraction $0 \leq p \leq 1$ of the smallest magnitude weights are pruned (layer-wise pruning). Then, after $k$ iterations of pruning, it holds that

$$4E_{AA}(H) \geq \sigma_W^2 + d\sigma_X^2\sigma_W^2\left((1-p)^k + \sqrt{\frac{4}{\pi}}\,\text{erf}^{-1}\left(1 - (1-p)^k\right)e^{-\left(\text{erf}^{-1}\left(1 - (1-p)^k\right)\right)^2}\right) \tag{1}$$

where $\text{erf}^{-1}(.)$ is the inverse error function.

Next, based on the above result, the following theorem establishes how the *average gradient norm* depends on the LR of the neural network, and the pruning iteration.

**Theorem 2.** In the setting of Theorem 1, we consider a single epoch of weight update for the network across a training dataset $S = \{(X_1, Y_1), .., (X_n, Y_n)\}$ using the cross-entropy loss, where $Y_i \in \{0, 1\}$. Let

$\alpha$ denote the learning rate. Let us denote the R.H.S of equation 1 by $C(\sigma_X, \sigma_W, p, k)$. Let the final layer weights before and after one training epoch be $W_2$ and $W_2'$ respectively. We have,

$$\mathop{\mathbb{E}}_{W_2 \sim \mathcal{N}_k(0, \sigma_W^2 I)} [E_{WG}(W_2, W_2')] \geq \alpha^2 \gamma C(\sigma_X, \sigma_W, p, k), \tag{2}$$

for some constant $\gamma$, where $\mathcal{N}_k(0, \sigma_w^2 I)$ represents the distribution of $W_2$ after being initialized as the Gaussian $\mathcal{N}(0, \sigma_w^2 I)$ and pruned for $k$ iterations.

**Remark 1. (Pruning and LR)** Theorems 1 and 2 together establish how the choice of LR influences the lower bound of *average gradient norm*. Theorem 1 shows that the lower bound of *activation norm* of the hidden layer decreases as the network is pruned, and as Theorem 2 shows, this also reduces the lower bound of *average gradient norm* per epoch. Thus, to counter this reduction, it is necessary to increase the learning rate $\alpha$ as the number of pruning cycles grows, in order to ensure that the R.H.S of equation 2 remains fixed.

**Remark 2. (S-shape of LR During Iterative Pruning)** We fix the average gradient norm to negate the impact of weight gradients reducing while pruning (see Fig. 1). Theorem 2 implies that to maintain a fixed *average gradient norm* of $\mathbb{E}_{W_2 \sim \mathcal{N}_k(0, \sigma_W^2 I)}[E_{WG}(W_2, W_2')] = K$, we must have the learning rate $\alpha \leq (K/\gamma C(\sigma_X, \sigma_W, p, k))^{1/2}$. We find that this upper bound resembles an S-shape trajectory during iterative pruning (see the red line in Fig. 2).

**Remark 3. (Additional Results)** Note that we extend Theorem 1's result to the case of fully connected neural networks of arbitrary depth in Proposition 1 of the Appendix. Similarly, we extend Theorem 2's result to the case of networks of arbitrary depth in Corollary 1 of the Appendix. Although Theorem 1 assumes 2-layer neural networks of infinite width, we extend it to the case of finite hidden neurons (see Proposition 2 of Appendix), yielding a probabilistic bound of the same form as Theorem 1. Note that the results in both Theorems are empirically verified in Section B of the Appendix.

## 4 A New Learning Rate Schedule

In Section 4.1, we shortlist four LR benchmarks for comparison. In Section 4.2, we introduce SILO and highlight the difference with existing works. In Section 4.3, we detail the algorithm for SILO.

### 4.1 LR Schedule Benchmarks

Learning rate is the most important hyperparameter in training neural networks (Goodfellow et al., 2016). The LR schedule is to adjust the value of LR during training by a pre-defined schedule. Three common LR schedules are summarized as follows.

1. **LR Decay** starts with a large LR and linearly decays it by a certain factor after a pre-defined number of epochs. Several recent works (You et al., 2019; Ge et al., 2019; An et al., 2017) have demonstrated that decaying LR helps the neural network to converge better and avoids undesired oscillations in optimization.

2. **LR Warmup** is to increase the LR to a large value over certain epochs and then decreases the LR by a certain factor. It is a popular schedule used by many practitioners for transfer learning (He et al., 2019) and network pruning (Frankle & Carbin, 2019; Frankle et al., 2020).

3. **Cyclical LR** (Smith, 2017) varies the LR cyclically between a pre-defined lower and upper bound. It has been widely used in many tasks (You et al., 2019).

**All the three LR schedules described above and constant LR** will be used as **benchmarks for performance comparison**. We note that, in addition to LR schedules which vary LR by a pre-defined schedule, adaptive LR optimizers such as AdaDelta (Zeiler, 2012) and Adam (Kingma & Ba, 2014) provide heuristic based approaches to adaptively vary the step size of weight update based on observed statistics of the past gradients. All of them are sophisticated optimization algorithms and much work (Gandikota et al., 2021; Jentzen et al., 2021) has been done to investigate their mechanisms. In this paper, the performance of all benchmarks and SILO will be evaluated using SGD with momentum = 0.9 and a weight decay of 1e-4 (same as (Renda et al., 2019; Frankle & Carbin, 2019)). The effect of those adaptive LR optimizers on SILO will be discussed in Section 6.

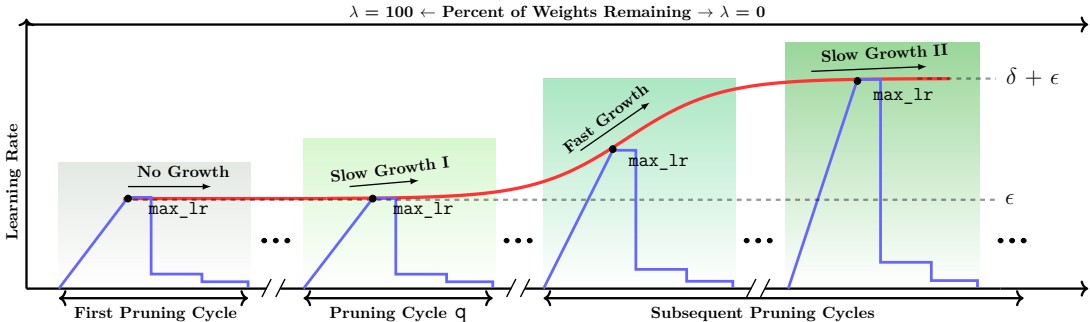

Figure 2: Illustration of SILO during pruning. The S-shape red line is motivated from Theorem 2.

## 4.2 SILO Learning Rate Schedule

To ensure the pruned network is properly trained during iterative pruning, we propose an S-shaped Improved Learning rate Schedule, called SILO, for iterative pruning of networks. As illustrated in Fig. 2, the main idea of the proposed SILO is to apply the LR warmup schedule at every pruning cycle, with a gradual increase of the LR upper bound (i.e., `max_lr`) in an S-shape as the network is iteratively pruned. This LR warmup schedule is meant to be flexible and can change depending on different networks and datasets.

The S-shape in SILO is inspired by Theorem 2 (see Remark 2) and will be further verified by comparing to an Oracle. We divide the S-shape into four phases and provide the intuition behind each phase as follows.

1. Phase-1: **No Growth**, SILO does not increase `max_lr` until the pruning cycle `q` (see Fig. 2). It is because the unpruned network often contains a certain amount of weights with zero magnitude. Those parameters are likely to be pruned at the first few pruning cycles, and removing such weights has negligible effect on the distribution of weight gradients.

2. Phase-2: **Slow Growth I**, the pruning algorithm has removed most zero magnitude weights and started pruning weights with small magnitude. Pruning such weights has a small effect on distribution of weight gradients. Hence, we slightly increase `max_lr` after pruning cycle `q`.

3. Phase-3: **Fast Growth**, SILO greatly increases `max_lr`. It is because the pruning algorithm now starts removing weights with large magnitude and the distribution of weight gradients becomes much narrower. This requires a much larger LR for meaningful weight updates.

4. Phase-4: **Slow Growth II**, the network is now heavily pruned and very few parameters left in the network. By using the same pruning rate, a very small portion of the weights will be pruned. This could cause a marginal effect on the distribution of weight gradients. Hence, SILO slightly increases `max_lr`.

We note that SILO is designed based on the assumption that existing pruning methods tend to prune weights with small magnitude. **The key difference with existing LR schedules** (e.g., LR warmup) is that SILO is adaptive and able to **precisely increase** the value of `max_lr` as the network is iteratively pruned, while existing LR schedules do not factor in the need to change `max_lr` during different pruning cycles.

## 4.3 Implementation of SILO

As for the implementation of SILO, we designed a function to estimate the value of `max_lr` as shown below.

$$\texttt{max\_lr} = \frac{\delta}{1 + (\frac{\gamma}{1-\gamma})^{-\beta}} + \epsilon, \tag{3}$$

where $\gamma = 1 - (1 - \texttt{p})^{m-\texttt{q}}$ is the input of the function and `max_lr` is the output of the function. When computing the value of $\gamma$, the parameter `p` is the pruning rate and `m` is the number of completed pruning cycles. In equation 3, the parameters $\beta$ and `q` are used to control the shape of the S curve. The larger the $\beta$, the later the curve enters the Fast Growth phase. The parameter `q` determines at which pruning cycle SILO enters the Slow Growth I phase. When $q = 0$, the No Growth phase will be skipped and $\gamma$ will be

---

**Algorithm 1** Algorithm for the proposed S-shaped Improved Learning rate Optimization (SILO)

---

**Input:** lower bound $\epsilon$, upper bound $\delta + \epsilon$, pruning rate $\mathtt{p}$, number of pruning cycles $\mathtt{L}$, number of training epochs $\mathtt{t}$, S-shape control term $\beta$, delay term $\mathtt{q}$.

1: **for** $\mathtt{m} = 0$ to $\mathtt{L}$ **do**
2:      **if** $\mathtt{m} \leq \mathtt{q}$ **then**
3:          $\mathtt{max\_lr} = \epsilon$
4:      **else**
5:          $\mathtt{max\_lr} = \frac{\delta}{1+(\frac{\gamma}{1-\gamma})^{-\beta}} + \epsilon$, $\gamma = 1 - (1 - \mathtt{p})^{\mathtt{m}-\mathtt{q}}$
6: **for** $\mathtt{i} = 0$ to $\mathtt{t}$ **do**
7:      (1) linearly warmup the LR to $\mathtt{max\_lr}$
8:      (2) drop the value of LR by 10 at certain epochs

---

the proportion of pruned weights at the current pruning cycle. The parameters $\epsilon$ and $\delta$ determine the value range of $\mathtt{max\_lr}$. As the network is iteratively pruned, $\gamma$ increases and $\mathtt{max\_lr}$ increases from $\epsilon$ to $\epsilon + \delta$ accordingly. The details of the SILO algorithm are summarized in Algorithm 1.

**Parameter Selection for SILO.** Algorithm 1 requires several inputs for implementation. The value of $\epsilon$ can be tuned using the validation accuracy of the unpruned network while the value of $\delta$ can be tuned using the validation accuracy of the pruned network with targeted sparsity. The pruning rate $\mathtt{p}$ and pruning cycles $\mathtt{L}$ are chosen to meet the target sparsity. The number of training epochs $\mathtt{t}$ should be large enough to guarantee the network convergence. Let $\mathtt{q} = 1$ and $\beta = 5$ could be a good choice and yield promising results as we demonstrate in Section 5. Furthermore, based on our experience, the value of $\mathtt{q}$ and $\beta$ could be tuned in the range of $[0, 3]$ and $[3, 6]$, respectively. Lastly, we note that the sensitivity of parameters in the proposed SILO will be further examined in Section 5.3.

## 5 Performance Evaluation

In Sections 5.1, we first summarize the experiment setup. In Section 5.2, compare the performance of the proposed SILO to four LR schedule benchmarks. In Section 5.3, we examine the sensitivity of parameters in SILO. In Section 5.4, we present the value of $\mathtt{max\_lr}$ estimated by the proposed SILO at each pruning cycle and compare it to an Oracle which exhaustively searches for the optimal $\mathtt{max\_lr}$ via grid search.

### 5.1 Experimental Setup

We demonstrate that SILO can work well with different pruning methods across a wide range of networks and datasets. We shortlist two classical pruning methods (global weight, global gradient) suggested by (Blalock et al., 2020) and three state-of-the-art pruning method (Iterative Magnitude Pruning (IMP) (Frankle & Carbin, 2019), Layer-adaptive Magnitude-based Pruning (LAMP) (Lee et al., 2020)) and Lookahead Pruning (LAP) (Park et al., 2020). The details for each experiment are as follows.

1. Pruning ResNet-20 (He et al., 2016) on CIFAR-10 via global magnitude.

2. Pruning VGG-19 (Simonyan & Zisserman, 2014) on CIFAR-10 via global gradient.

3. Pruning DenseNet-40 (Huang et al., 2017) on the CIFAR-100 dataset using LAMP.

4. Pruning MobileNetV2 (Sandler et al., 2018) on the CIFAR-100 dataset using LAP.

5. Pruning ResNet-50 on ImageNet (i.e., ImageNet-1000) (Deng et al., 2009) using IMP.

6. Pruning Vision Transformer (ViT-B-16) (Dosovitskiy et al., 2020) on CIFAR-10 using IMP.

In each experiment, **to demonstrate the robustness of parameters in the proposed SILO, we compare SILO with a fixed value of ($\mathtt{q} = 1$, $\beta = 5$) to constant LR and the three shortlisted LR schedules**: (i) LR decay, (ii) cyclical LR and (iii) LR warmup (described in Section 4.1). The implementation details of each LR schedule are summarized in Table 1.

| Schedule | Description (Iters: Iterations) |
|---|---|
| LR decay (`a`, `b`) | linearly decay the value of LR from `a` over `b` Iters. |
| cyclical LR (`a`, `b`, `c`) | linearly vary between `a` and `b` with a step size of `c` Iters. |
| LR warmup (`a`, `b`, `c`, `d`, `e`) | increase to `a` over `b` Iters, 10x drop at `c`, `d`, `e` Iters. |
| SILO ($\epsilon$, $\delta$, `b`, `c`, `d`, `e`) | LR warmup (`max_lr`, `b`, `c`, `d`, `e`), where `max_lr` increases from $\epsilon$ to $\epsilon + \delta$ during iterative pruning (see equation 3). |

Table 1: Descriptions of LR schedule benchmarks and the proposed SILO.

**(1) Methodology.** We train the network using the training dataset via SGD with momentum = 0.9 and a weight decay of 1`e`-4 (same as (Renda et al., 2019; Frankle & Carbin, 2019)). Next, we prune the trained network with a pruning rate of 0.2 (i.e., 20% of remaining weights are pruned) in 1 pruning cycle. We repeat 25 pruning cycles in 1 run and use early-stop top-1 test accuracy (i.e., the corresponding test accuracy when early stopping criteria for validation error is met) to evaluate the performance. The results are averaged over 5 runs and the corresponding standard deviation are summarized in Tables 2 - 7, where the results of pruning ResNet-20, VGG-19, DenseNet-40, MobileNetV2, ResNet-50 and Vision Transformer (ViT-B-16) are shown, respectively. **Some additional details (e.g., training epochs, optimizer, batch size, etc) and results for more values of $\lambda$ are given in Tables 16 - 21 in the Appendix**.

**(2) SOTA LR Schedules.** To ensure fair comparison against prior SOTA LR schedules, we use LR schedules reported in the literature. Specifically, LR schedules (i.e., LR-warmup) from Table 2 - 6 are from (Frankle & Carbin, 2019), (Frankle et al., 2020), (Zhao et al., 2019), (Chin et al., 2020) and (Renda et al., 2019), respectively. The LR schedule (i.e., cosine decay) in Table 7 is from (Dosovitskiy et al., 2020).

**(3) Parameters for other LR schedules.** For the other schedules without a single "best" LR in the literature, we tune the value of LR for each of them via a grid search with a range from 1`e`-4 to 1`e`-1 using the validation accuracy. Other related parameters (e.g., step size) are also tuned in the same manner. Lastly, we highlight that all LR schedules used, including SILO, are rewound to the initial state at the beginning of each pruning cycle, which is the same as the LR rewinding in (Renda et al., 2019).

**(4) Source Code & Devices:** We use Tesla V100 devices for our experiments, and the source code is available at `https://github.com/Martin1937/SILO`.

### 5.2 Performance Comparison

**(1) Reproducing SOTA results.** By using the implementations reported in the literature, we have correctly reproduced SOTA results. For example, the benchmark results of LR warmup in our Tables 2 - 7 are comparable to Fig. 11 and Fig. 9 of (Blalock et al., 2020), Table 4 in (Liu et al., 2019b), Fig. 3 in (Chin et al., 2020), Fig. 10 in (Frankle et al., 2020), Table 5 in (Dosovitskiy et al., 2020), respectively.

**(2) SILO outperforms SOTA results.** The key innovation of SILO is that the LR precisely increases as the network is pruned, by increasing `max_lr` in an S-shape as $\lambda$ decreases. This results in a much higher accuracy than all LR schedule benchmarks studied. For example, in Table 2, the top-1 test accuracy of SILO is 1.8% higher than the best performing schedule (i.e., LR-warmup) at $\lambda = 5.72$. SILO also obtains the best performance when using larger models in Table 3 (i.e., 2.6% higher at $\lambda = 5.72$) and using more difficult datasets in Table 4 (i.e., 4.0% higher at $\lambda = 5.72$).

**(3) Performance on ImageNet.** In Table 6, we show the performance of SILO using IMP (i.e., the lottery ticket hypothesis pruning method) via ResNet-50 on ImageNet which contains over 1.2 million images from 1000 different classes. We observe that SILO still outperforms the best performing LR schedule benchmark (LR-warmup) by 1.9% at $\lambda = 8.59$. This improvement increases to 3.2% when $\lambda$ reduces to 5.72.

**(4) Performance on SOTA networks (Vision Transformer).** Several recent works (Liu et al., 2021; Yuan et al., 2021) demonstrated that transformer based networks tend to provide excellent performance

| Original Top-1 Test Accuracy = 91.7% ($\lambda = 100$) | | | |
|---|---|---|---|
| $\lambda$ | 32.8 | 26.2 | 8.59 | 5.72 |
| constant LR | $88.1 \pm 0.9$ | $87.5 \pm 0.7$ | $82.8 \pm 0.9$ | $79.1 \pm 0.8$ |
| LR decay | $89.8 \pm 0.4$ | $89.0 \pm 0.7$ | $83.9 \pm 0.6$ | $79.8 \pm 0.7$ |
| cyclical LR | $89.7 \pm 0.6$ | $88.2 \pm 0.7$ | $84.1 \pm 0.8$ | $80.3 \pm 0.7$ |
| LR-warmup | $90.3 \pm 0.4$ | $89.8 \pm 0.6$ | $85.9 \pm 0.9$ | $81.2 \pm 1.1$ |
| SILO (Ours) | $\mathbf{90.8} \pm \mathbf{0.5}$ | $\mathbf{90.3} \pm \mathbf{0.4}$ | $\mathbf{87.5} \pm \mathbf{0.8}$ | $\mathbf{82.7} \pm \mathbf{1.2}$ |

Table 2: Top-1 test accuracy $\pm$ standard deviation of pruning ResNet-20 on CIFAR-10 via global magnitude.

| Original Top-1 Test Accuracy = 92.2% ($\lambda = 100$) | | | |
|---|---|---|---|
| $\lambda$ | 32.8 | 26.2 | 8.59 | 5.72 |
| constant LR | $88.8 \pm 0.6$ | $87.4 \pm 0.7$ | $82.2 \pm 1.4$ | $73.7 \pm 1.3$ |
| LR decay | $89.4 \pm 0.4$ | $88.6 \pm 0.5$ | $83.3 \pm 0.8$ | $75.4 \pm 0.9$ |
| cyclical LR | $89.8 \pm 0.5$ | $89.1 \pm 0.6$ | $83.7 \pm 1.0$ | $75.7 \pm 1.2$ |
| LR-warmup | $90.2 \pm 0.5$ | $89.8 \pm 0.8$ | $84.5 \pm 0.9$ | $76.5 \pm 1.0$ |
| SILO (Ours) | $\mathbf{90.6} \pm \mathbf{0.6}$ | $\mathbf{90.3} \pm \mathbf{0.6}$ | $\mathbf{86.1} \pm \mathbf{0.8}$ | $\mathbf{78.5} \pm \mathbf{1.0}$ |

Table 3: Top-1 test accuracy $\pm$ standard deviation of pruning VGG-19 on CIFAR-10 using global gradient.

| Original Top-1 Test Accuracy = 74.6% ($\lambda = 100$) | | | |
|---|---|---|---|
| $\lambda$ | 32.8 | 26.2 | 8.59 | 5.72 |
| constant LR | $70.3 \pm 0.8$ | $68.1 \pm 0.7$ | $60.8 \pm 1.1$ | $59.1 \pm 1.2$ |
| LR decay | $71.2 \pm 0.8$ | $69.0 \pm 0.6$ | $62.6 \pm 1.2$ | $60.3 \pm 1.4$ |
| cyclical LR | $70.9 \pm 0.6$ | $69.4 \pm 0.6$ | $63.0 \pm 1.1$ | $60.8 \pm 1.3$ |
| LR-warmup | $71.5 \pm 0.7$ | $69.6 \pm 0.8$ | $63.9 \pm 1.0$ | $61.2 \pm 0.9$ |
| SILO (Ours) | $\mathbf{72.4} \pm \mathbf{0.7}$ | $\mathbf{70.8} \pm \mathbf{0.8}$ | $\mathbf{65.7} \pm \mathbf{1.2}$ | $\mathbf{63.7} \pm \mathbf{1.0}$ |

Table 4: Top-1 test accuracy $\pm$ standard deviation of pruning DenseNet-40 on CIFAR-100 using LAMP.

| Original Top-1 Test Accuracy = 73.7% ($\lambda = 100$) | | | |
|---|---|---|---|
| $\lambda$ | 32.8 | 26.2 | 8.59 | 5.72 |
| constant LR | $69.8 \pm 1.1$ | $68.2 \pm 0.9$ | $63.8 \pm 1.1$ | $62.1 \pm 1.2$ |
| LR decay | $70.9 \pm 1.0$ | $69.4 \pm 0.6$ | $65.1 \pm 0.8$ | $64.0 \pm 1.1$ |
| cyclical LR | $71.5 \pm 0.7$ | $69.6 \pm 0.6$ | $65.3 \pm 1.1$ | $64.3 \pm 1.2$ |
| LR-warmup | $72.1 \pm 0.8$ | $70.5 \pm 0.9$ | $66.2 \pm 1.1$ | $64.8 \pm 1.5$ |
| SILO (Ours) | $\mathbf{72.5} \pm \mathbf{0.6}$ | $\mathbf{71.0} \pm \mathbf{0.7}$ | $\mathbf{68.8} \pm \mathbf{0.8}$ | $\mathbf{66.8} \pm \mathbf{1.4}$ |

Table 5: Top-1 test accuracy $\pm$ standard deviation of pruning MobileNetV2 on CIFAR-100 using LAP.

in computer vision tasks (e.g., classification). We now examine the performance of SILO using Vision Transformer (i.e., ViT-B16 with a resolution of 384). We note that the ViT-B16 uses Gaussian Error Linear Units (GELU, GELU(x) = $x\Phi(x)$, where $\Phi(x)$ is the standard Gaussian cumulative distribution function) as the activation function. We note that both ReLU and GELU have the unbounded output, suggesting that SILO could be helpful for pruning GELU based models as well.

We repeat the same experiment setup as above and compare the performance of SILO to other LR schedules using ViT-B16 in Table 7. We observe that SILO is able to outperform the standard implementation (cosine decay, i.e., decay the learning rate via the cosine function) by 1.3% at $\lambda = 8.59$ in top-1 test accuracy. This improvement increases to 1.6% when $\lambda$ reduces to 5.72.

| Original Top-1 Test Accuracy = 77.0% ($\lambda = 100$) | | | | |
|---|---|---|---|---|
| $\lambda$ | 32.8 | 26.2 | 8.59 | 5.72 |
| constant LR | $74.2 \pm 0.8$ | $73.9 \pm 0.7$ | $70.5 \pm 0.6$ | $69.2 \pm 0.9$ |
| LR decay | $75.6 \pm 0.5$ | $75.1 \pm 0.5$ | $72.7 \pm 0.8$ | $70.5 \pm 0.6$ |
| cyclical LR | $76.5 \pm 0.5$ | $75.5 \pm 0.6$ | $73.4 \pm 0.8$ | $71.2 \pm 0.7$ |
| LR-warmup | $76.6 \pm 0.2$ | $75.8 \pm 0.3$ | $73.8 \pm 0.5$ | $71.5 \pm 0.4$ |
| SILO (Ours) | $\mathbf{76.8} \pm \mathbf{0.4}$ | $\mathbf{76.1} \pm \mathbf{0.7}$ | $\mathbf{75.2} \pm \mathbf{0.8}$ | $\mathbf{73.8} \pm \mathbf{0.6}$ |

Table 6: Top-1 test accuracy $\pm$ standard deviation of pruning ResNet-50 on ImageNet using IMP.

| Original Top-1 Test Accuracy = 98.0% ($\lambda = 100$) | | | | |
|---|---|---|---|---|
| $\lambda$ | 32.8 | 26.2 | 8.59 | 5.72 |
| constant LR | $96.4 \pm 0.5$ | $96.0 \pm 0.7$ | $83.0 \pm 0.9$ | $80.1 \pm 0.8$ |
| cosine decay | $97.2 \pm 0.2$ | $96.5 \pm 0.6$ | $84.1 \pm 1.0$ | $81.6 \pm 1.1$ |
| cyclical LR | $97.0 \pm 0.2$ | $96.5 \pm 0.6$ | $83.4 \pm 0.6$ | $81.0 \pm 1.1$ |
| LR-warmup | $97.3 \pm 0.6$ | $96.8 \pm 0.7$ | $84.4 \pm 0.8$ | $82.1 \pm 0.9$ |
| SILO (Ours) | $\mathbf{97.7} \pm \mathbf{0.5}$ | $\mathbf{97.4} \pm \mathbf{0.6}$ | $\mathbf{85.5} \pm \mathbf{0.9}$ | $\mathbf{83.4} \pm \mathbf{0.8}$ |

Table 7: Top-1 test accuracy $\pm$ standard deviation of pruning Vision Transformer on CIFAR-10 using IMP.

| Percent of Weights Remaining, $\lambda$ | 32.8 | 26.2 | 8.59 | 5.72 |
|---|---|---|---|---|
| LR-warmup (benchmark) | $90.3 \pm 0.4$ | $89.8 \pm 0.6$ | $85.9 \pm 0.9$ | $81.2 \pm 1.2$ |
| SILO (q = 1, $\beta = 5$) | $90.8 \pm 0.5$ | $90.3 \pm 0.4$ | $87.5 \pm 0.8$ | $82.7 \pm 1.2$ |
| SILO (q = 2, $\beta = 5$) | $90.5 \pm 0.6$ | $90.1 \pm 0.2$ | $86.9 \pm 0.4$ | $82.4 \pm 1.1$ |
| SILO (q = 3, $\beta = 4$) | $90.9 \pm 0.4$ | $90.2 \pm 0.5$ | $87.1 \pm 0.9$ | $82.2 \pm 1.4$ |
| SILO (q = 1, $\beta = 3$) | $90.5 \pm 0.9$ | $90.6 \pm 0.8$ | $87.5 \pm 0.6$ | $83.1 \pm 1.6$ |
| SILO (q = 1, $\beta = 7$) | $90.4 \pm 0.2$ | $89.2 \pm 0.7$ | $85.1 \pm 1.2$ | $80.8 \pm 1.9$ |
| SILO (q = 5, $\beta = 5$) | $90.2 \pm 0.3$ | $89.8 \pm 0.5$ | $85.5 \pm 1.0$ | $80.5 \pm 1.6$ |

Table 8: Performance (Top-1 test accuracy $\pm$ standard deviation) of SILO with different values of $q$ and $\beta$ when pruning ResNet-20 on CIFAR-10 via global magnitude.

| Percent of Weights Remaining, $\lambda$ | 32.8 | 26.2 | 8.59 | 5.72 |
|---|---|---|---|---|
| LR-warmup (benchmark) | $90.2 \pm 0.5$ | $89.8 \pm 0.8$ | $84.5 \pm 0.9$ | $76.5 \pm 1.0$ |
| SILO (q = 1, $\beta = 5$) | $90.6 \pm 0.6$ | $90.3 \pm 0.6$ | $86.1 \pm 0.8$ | $78.5 \pm 1.0$ |
| SILO (q = 2, $\beta = 5$) | $90.7 \pm 0.5$ | $90.0 \pm 0.4$ | $86.5 \pm 0.9$ | $79.2 \pm 1.3$ |
| SILO (q = 3, $\beta = 4$) | $90.3 \pm 0.2$ | $90.4 \pm 0.3$ | $85.8 \pm 0.6$ | $78.8 \pm 0.9$ |
| SILO (q = 1, $\beta = 3$) | $90.8 \pm 0.8$ | $90.1 \pm 0.6$ | $85.5 \pm 0.9$ | $78.2 \pm 1.1$ |
| SILO (q = 1, $\beta = 7$) | $90.0 \pm 0.4$ | $88.5 \pm 0.9$ | $82.1 \pm 1.4$ | $75.3 \pm 1.3$ |
| SILO (q = 5, $\beta = 5$) | $90.1 \pm 0.5$ | $89.1 \pm 0.6$ | $83.2 \pm 1.1$ | $75.7 \pm 1.5$ |

Table 9: Performance (Top-1 test accuracy $\pm$ standard deviation) of SILO with different values of $q$ and $\beta$ when pruning VGG-19 on CIFAR-10 via global gradient.

## 5.3 Sensitivity of Parameters in SILO

In Section 5.2, we demonstrate the robustness of parameters in SILO by using a fixed value of q = 1, $\beta = 5$ and compare it to benchmark schedulers using different networks, datasets and pruning methods. We now repeat the setup in Tables 2 - 3 and examine the sensitivity of these two parameters in Tables 8 - 9.

| Percent of Weights Remaining, $\lambda$ | 100 | 51.3 | 32.9 | 21.1 | 5.72 |
|---|---|---|---|---|---|
| Oracle optimized `max_lr` ($\times 10^{-2}$) | 4 | 4.6 | 9.0 | 9.8 | 10.2 |
| Oracle optimized interval ($\times 10^{-2}$) | [3.6, 4.2] | [4.2, 5.4] | [8.0, 9.6] | [9.2, 10.4] | [9.8, 10.6] |
| SILO's estimated `max_lr` ($\times 10^{-2}$) | 4 | 4.32 | 9.2 | 9.9 | 9.99 |

Table 10: Comparison between Oracle optimized `max_lr`, Oracle optimized interval (both obtained via grid search) and the value of `max_lr` estimated by SILO when iteratively pruning VGG-19 on CIFAR-10.

| Percent of Weights Remaining, $\lambda$ | 100 | 51.3 | 41.1 | 32.9 | 21.1 |
|---|---|---|---|---|---|
| Oracle optimized `max_lr` ($\times 10^{-2}$) | 3.4 | 3.8 | 4.6 | 5.6 | 6.2 |
| Oracle optimized interval ($\times 10^{-2}$) | [2.8, 3.6] | [3.4, 4.2] | [3.8, 5.2] | [5.4, 6.6] | [5.4, 6.8] |
| SILO estimated `max_lr` ($\times 10^{-2}$) | 3 | 3.2 | 4.7 | 6.4 | 6.9 |

Table 11: Comparison between Oracle tuned `max_lr`, Oracle optimized `max_lr` interval (both obtained via grid search) and `max_lr` estimated by SILO when iteratively pruning ResNet-20 on the CIFAR-10 dataset.

The takeaway message from Tables 8 - 9 is three-fold: (i) The parameters of SILO can provide promising results (i.e., outperform benchmark schedulers) within the suggested value range, i.e., $0 \leq q \leq 3, 3 \leq \beta \leq 6$ (see first 5 rows in Tables 8 - 9). (ii) When the value of $q$ and $\beta$ fall outside the suggested range, SILO may fail to outperform benchmarks (see last 2 rows in Tables 8 - 9). (iii) It also suggests that $q$ and $\beta$ could be tuned within a relatively small range and does not require as much effort as tuning the hyperparameters.

### 5.4 Comparing SILO to an Oracle

Our new insight suggests that, due to the change in distribution of hidden representations during iterative pruning, LR should be re-tuned at each pruning cycle. SILO provides a method to adjust the `max_lr` in an S-shape, which is backed up by a theoretical result (see Theorem 2). We now further examine the S-shape trajectory of SILO by comparing SILO's estimated `max_lr` to an Oracle, which uses the same LR warmup structure as SILO but exhaustively searches for the optimal value of `max_lr` at each pruning cycle. The Oracle's `max_lr` at the current pruning cycle is chosen by grid search ranging from `1e-4` to `1e-1` and the best performing value (i.e., determined by validation accuracy) is used to train the network. The results of `max_lr` determined this way when iteratively pruning a VGG-19 on CIFAR-10 using the global magnitude are detailed in Table 10 via two metrics:

1. **Oracle optimized `max_lr`**: The value of `max_lr` that provides the best validation accuracy.

2. **Oracle optimized interval**: The range of `max_lr` which provides comparable performance to Oracle optimized `max_lr` (i.e., within 0.5% of the best validation accuracy).

**SILO vs Oracle (Performance):** In Table 10, the `max_lr` estimated by SILO falls in the Oracle optimized `max_lr` interval at each pruning cycle, suggesting that SILO can precisely adjust `max_lr` to provide competitive performance with Oracle. This further verifies the S-shape trajectory of `max_lr` used in SILO.

**SILO vs Oracle (Complexity):** The process of finding the Oracle tuned `max_lr` requires a significantly larger computational complexity in tuning due to the grid search. Assume that `max_lr` is searched from a sampling space of $[\theta_1, \cdots, \theta_n]$ for $k$ pruning cycles. Hence, the complexity of the Oracle will be $\mathcal{O}(n^k)$. On the other hand, SILO controls the variation of `max_lr` at each pruning cycle via four parameters: ranges of `max_lr`: $[\epsilon, \epsilon + \delta]$, delay term `q` and S-shape control term $\beta$. Similar to the Oracle, both $\epsilon$ and $\delta$ can be searched from a range of $n$ values. As we have recommended before, `q` and $\beta$ can be tuned in the range of $[0, 3], [3, 6]$, respectively. As a result, SILO has a complexity of $\mathcal{O}(n^2)$, which is exponentially less complex than the Oracle's complexity, but with competitive performance. Lastly, we highlight that similar performance trends can be observed using ResNet-20 via global magnitude (see Table 11).

| **Params**: 227K; **Train Steps**: 63K Iters; **Batch**: 128; **Pruning Rate**: 0.2 | | | | |
|---|---|---|---|---|
| $\lambda$ | 100 | 32.9 | 21.1 | 5.72 | 2.03 |
| constant LR | $88.4\pm_{0.4}$ | $84.8\pm_{0.6}$ | $83.5\pm_{0.6}$ | $75.5\pm_{1.2}$ | $67.1\pm_{1.7}$ |
| LR decay | $88.6\pm_{0.3}$ | $87.1\pm_{0.7}$ | $83.7\pm_{0.9}$ | $76.1\pm_{0.8}$ | $66.0\pm_{1.3}$ |
| cyclical LR | $88.9\pm_{0.3}$ | $86.9\pm_{0.5}$ | $84.1\pm_{0.3}$ | $77.0\pm_{0.9}$ | $64.4\pm_{1.1}$ |
| LR-warmup | $89.1\pm_{0.3}$ | $87.2\pm_{0.4}$ | $84.5\pm_{0.6}$ | $75.2\pm_{1.1}$ | $65.1\pm_{1.9}$ |
| SILO | $\mathbf{89.2}\pm_{\mathbf{0.2}}$ | $\mathbf{87.9}\pm_{\mathbf{0.3}}$ | $\mathbf{86.3}\pm_{\mathbf{0.5}}$ | $\mathbf{79.5}\pm_{\mathbf{1.7}}$ | $\mathbf{71.7}\pm_{\mathbf{2.3}}$ |

Table 12: Performance comparison (averaged top-1 test accuracy $\pm$ std over 5 runs) of iteratively pruning ResNet-20 on CIFAR-10 dataset using global magnitude and Adam optimizer (Kingma & Ba, 2014).

| **Params**: 227K; **Train Steps**: 63K Iters; **Batch**: 128; **Pruning Rate**: 0.2 | | | | |
|---|---|---|---|---|
| $\lambda$ | 100 | 32.9 | 21.1 | 5.72 | 2.03 |
| constant LR | $87.9\pm_{0.3}$ | $83.4\pm_{0.4}$ | $81.5\pm_{0.9}$ | $65.5\pm_{1.9}$ | $55.1\pm_{2.3}$ |
| LR decay | $88.4\pm_{0.2}$ | $84.8\pm_{0.6}$ | $77.8\pm_{0.9}$ | $67.1\pm_{1.4}$ | $58.3\pm_{1.6}$ |
| cyclical LR | $88.1\pm_{0.3}$ | $84.7\pm_{0.5}$ | $81.9\pm_{0.7}$ | $67.5\pm_{0.9}$ | $56.3\pm_{1.7}$ |
| LR-warmup | $\mathbf{88.9}\pm_{\mathbf{0.2}}$ | $85.1\pm_{0.5}$ | $81.7\pm_{0.4}$ | $67.3\pm_{1.3}$ | $57.1\pm_{1.4}$ |
| SILO | $88.7\pm_{0.3}$ | $\mathbf{86.1}\pm_{\mathbf{0.4}}$ | $\mathbf{83.1}\pm_{\mathbf{0.6}}$ | $\mathbf{72.5}\pm_{\mathbf{1.3}}$ | $\mathbf{63.5}\pm_{\mathbf{1.9}}$ |

Table 13: Performance comparison (top-1 test accuracy $\pm$ std over 5 runs) of iteratively pruning ResNet-20 on CIFAR-10 dataset using global magnitude and RMSProp optimizer (Tieleman & Hinton, 2012).

## 6 Reflections

In summary, SILO is an adaptive LR schedule for network pruning with theoretical justification. SILO outperforms existing benchmarks by 2.1% - 3.2% via classical networks and datasets. For SOTA networks (e.g., Vision Transformer) and large scale datasets (e.g., ImageNet), it leads to an improvement of 3% - 5.6%. More importantly, via the S-shape trajectory, SILO can obtain comparable performance to Oracle with significantly lower complexity. We now conclude the paper by presenting some reflections.

**(1) The gain of SILO.** As compared to existing LR schedules, the advantage of SILO is three-fold: (i) SILO is specially designed for network pruning which gradually increases the value of LR in an S-shape as the network is gradually pruned. (ii) The S-shape trajectory is theoretically justified and empirically examined by comparing to an Oracle. The results suggest that SILO can obtain a comparable performance to the Oracle with a significantly lower complexity. (iii) As for its performance, SILO outperforms existing LR schedules by 2% - 4%, which is demonstrated via extensive experiments.

**(2) Connection to Prior Work.** Our work explores the important role of LR in network pruning and provides a new insight – *as the ReLU-based network is iteratively pruned, a larger LR should be used to retrain the pruned network*. This new insight could be used to explain the surprising effect of LR schedules observed in prior works. Specifically, Frankle & Carbin (2019) highlighted that they can only find winning tickets after applying a LR warmup schedule. Using insights from our analysis, we attribute this to LR warmup increasing the LR to a large value (e.g., Frankle & Carbin (2019) increases LR to 1e-1 when training VGG-19) which is better for pruned networks. Similarly, Renda et al. (2019) proposed LR rewinding and demonstrated that it outperforms standard fine-tuning. We attribute this to LR rewinding ensuring that a relatively larger LR (as compared to fine-tuning) is used to better train pruned networks in every pruning cycle.

**(3) Performance using Adaptive LR Optimizers.** In the main paper, we only evaluate the performance of SILO using SGD. We note that the weight update mechanism is different for other adaptive learning rate optimizers, which may potentially affect the performance of SILO. In Tables 12 - 13, we conduct a similar performance comparison using Adam (Kingma & Ba, 2014) and RMSprop (Tieleman & Hinton, 2012), and SILO still outperforms all LR schedule benchmarks studied.

| Percent of Weights Remaining, $\lambda$ | 30.4 | 24.7 | 7.29 |
|---|---|---|---|
| (i) LR-Warmup | $88.5 \pm 0.9$ | $87.1 \pm 1.2$ | $82.2 \pm 1.4$ |
| (ii) SILO (q = 1, $\beta = 3$) | $89.4 \pm 0.8$ | $88.5 \pm 1.4$ | $83.9 \pm 1.7$ |
| (iii) LR-Warmup | $87.4 \pm 0.7$ | $85.2 \pm 0.9$ | $80.8 \pm 1.6$ |
| (iv) SILO (q = 1, $\beta = 3$) | $88.1 \pm 0.6$ | $87.1 \pm 1.1$ | $82.3 \pm 1.4$ |

Table 14: Performance comparison between SILO and LR-warmup when pruning VGG-19 (rows (i) and (ii)) and ResNet-20 (rows (iii) and (iv)) using L1 Norm filter pruning (Li et al., 2017) on CIFAR-10 dataset.

**(4) Extension of SILO to Structured Pruning.** In Section 3.1, we have examined the distribution of weight gradients using structured pruning and shown that structured pruning also suffer from the issue of decreasing weight gradients (see Fig. 7 in the Appendix for more details). This suggests that SILO could be applicable to structured pruning as well.

In Table 14, we compare the performance of SILO to LR-warmup using L1 Norm filter pruning (Li et al., 2017) on ResNet-20 and VGG-19. We observe that SILO still outperforms the benchmark LR schedule, leading to an improvement of 2.0% (compare SILO at row (ii) to LR-warmup at row (i) when $\lambda = 7.29$). We note that the performance improvement is not as significant as that of working with unstructured pruning. We suspect it is due to that the distribution of weight gradients may change differently when the network is structurely pruned. In such a case, SILO may need to be re-designed for a larger performance gain.

**(5) Effect of SILO on Weight Update and Generalization.** We also examine the effect of SILO on distribution of weight update (weight gradient $\times$ learning rate) and the experimental results demonstrate that, with SILO, the distribution of weight update of pruned networks becomes less centralized, suggesting that SILO helps to mitigate the issue of decreasing weight gradients. As a result, the pruned network is better trained with SILO, leading to a much better generalization performance. We refer the interested reader to Figs. 8 - 10 in the Appendix.

**(6) Future Research.** (i) We demonstrate the performance of SILO on ReLU-based networks (ResNet, VGG) and GELU-based networks (Vision Transformer). We note that similar LR schedules could be used for networks with other activation functions (e.g., PReLU) and we plan to explore this in future research. (ii) Moreover, the main motivation for SILO is that the distribution of weight gradients tends to become narrower after pruning. An approach to automatically determine the value of `max_lr` from the distribution of weight gradients is definitely worth deeper thought. (iv) As mentioned above, when it comes to the case of structured pruning, the improvement of SILO is not as large as that of working with unstructured pruning. This could be due to that the distribution of weight gradients may change differently when the network is structurely pruned. In such a case, SILO may need to be re-designed for a larger performance gain.

## Acknowledgements

This research is supported by A*STAR, CISCO Systems (USA) Pte. Ltd and the National University of Singapore under its Cisco-NUS Accelerated Digital Economy Corporate Laboratory (Award I21001E0002). Additionally, we would like to thank the members of the Kent-Ridge AI research group at the National University of Singapore for helpful feedback and interesting discussions on this work.

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
