# OpenReview forum: "Optimizing Learning Rate Schedules for Iterative Pruning of Deep Neural Networks"
_TMLR — Accepted by TMLR_

### Review · Reviewer_d58D · 2023-05-30

**Summary Of Contributions:**

The paper suggests a new learning rate scheduler for iteratively pruning of deep ReLU networks. Motivated by the insight that the gradient magnitude, as well as the hidden representation, reduce during iterative pruning which the authors try to tackle and therefore motivate a new scheduler increasing the learning rate during the iterations. The method is well benchmarked against related works and baselines  and achieves SOTA results on various benchmarks.

**Audience:**

Yes

**Claims And Evidence:**

Yes

**Requested Changes:**

I think a couple of additional details and ablations would make the paper even stronger:

1. Does the method actually help increase the gradients, after scaling with a larger learning rate, and keep it constant throughout the cycles? This is just a verification that your motivation that mitigates this problem helps throughout pruning cycles makes sense.

2. Regarding your theoretical results: Can you verify these bounds in practice? If you apply pruning techniques in two-layer nets, do you observe these bounds in practice? And then, does your silo scheduler mitigates this in practice? This would be of great interest to me. Otherwise, these results seem a bit overwhelming although I admire the effort in producing these results. Disclaimer: I did not study the proofs of the results and therefore did not understand the implications from them fully. Please correct me if I misunderstood something.

3. Ablations: It would be nice to get an intuition of the sensitivity of the algorithms' hyperparameters. Can you provide a small study on one of your benchmarks that show what happens when moving e.g. \epsilon or \gamma?



**Strengths And Weaknesses:**

I enjoyed reading the paper as it was clearly written and I felt no details were hidden or missed. I also enjoyed the rather simple insights that lead motivates the theoretical and empirical study. The method has a few moving bits and pieces and a couple of hyperparameters that require tuning, nevertheless, the method seems quite robust. Although it wouldn't be a method that would be on the top of my list to try in practice, due to its additional complexity, I think the study is well executed and is of interest to the TMLR community.

---

> ### Author Response · Authors · 2023-06-20
> **Author Responses to Reviewer d58D (1/2)**
>
> Dear Reviewer,
>
> We are greatly thankful for your professional and prompt review work. Your insightful comments are highly appreciated. In the following, we outline changes to be made based on your suggestions. **We note that all clarifications and new results mentioned below will be updated in the revised version**.
>
> $\newline$
>
> **(1) [Effect of SILO on weight update]** Yes, the proposed SILO helps to increase the weight update (weight gradient $\times$ learning rate) and mitigates the issue of decreasing weight gradient during pruning.
>
> We repeat the experimental setup in Section 3.1 and compare the distribution of weight update without SILO and with SILO. Please see our plots in the following link.
>
> https://drive.google.com/file/d/1yEcyMcPy8YrLSfejP5TxaUKADcbK6UEo/view?usp=sharing
>
> We observe that, when we apply SILO,  the distribution of weight updates becomes less centralized and is closer to that of the unpruned network (i.e., compare pruned network ( $\lambda$ = 13.4) in Figure (a) to its counterpart in Figure (b)). This verifies that the proposed SILO helps to mitigate the issue of decreasing weight gradient during pruning.
>
> $\newline$
>
>
> **(2) [Verification of bounds in theoretical results]** Yes, we are able to verify the bounds in practice for Theorem 1 and Theorem 2. Further details regarding the conducted experiments that follow will be added to the revised version.
>
> For Theorem 1, we compute the average activation energy for a range of two layered neural network configurations with 1000 hidden nodes, 100 input dimensionality, on a toy classification problem where the ground truth function is known. For convenience, we repost the equation of Theorem 1 below
>
> $4E_{AA}(H) \geq \sigma_W^2 + d\sigma_X^2\sigma_W^2 \Big((1-p)^k +  \sqrt{\frac{4}{\pi}}{erf}^{-1}\Big(1-(1-p)^k\Big)e^{-\left({erf}^{-1}\left(1-(1-p)^k\right)\right)^2}\Big). $   (1)
>
> We find that in each case, the L.H.S of equation (1) is always greater than the R.H.S. Furthermore, as we increase the pruning percentage, we find that both the L.H.S and R.H.S of (1) get smaller and closer to each other. Moreover, in each case, we find that the trend of the R.H.S of (1) quite accurately reflects the trend of the empirically observed average activation energy. The plot in the following link depicts the observed average activation energy and its bound.
>
> https://drive.google.com/file/d/1PC9bWpozD1BFyy0rw3QP45_qVusEMlX6/view?usp=sharing
>
>
> $\newline$
>
> For Theorem 2, we first display the equation of Theorem 2 below for convenience
>
> $\mathbb{E}\left[E_{WG}\left(W_2,W_2'\right)\right] \geq \alpha^2\gamma C(\sigma_X,\sigma_W,p,k).  $  (2)
>
> We test how the average weight gradient energy changes as the $C(\sigma_X,\sigma_W,p,k)$ term in the R.H.S of (2) changes. Following the same setup as in the experiments of Theorem 1, we change the overall pruning percentage across a fixed range to change the R.H.S, and observe the changes in the average weight gradient energy. The 2-layer network is updated after one iteration of gradient descent using a cross-entropy loss, and the average weight gradient energy is computed by repeating the update iteration across 500 trials and then, we average the weight gradient energy across all trials. Importantly, the learning rate is fixed for all experiments, and thus, it does not depend on the number of pruning iterations $k$. We found that the changes in $C(\sigma_X,\sigma_W,p,k)$ very accurately reflect the changes in $E_{WG}$. Specifically, as $C(\sigma_X,\sigma_W,p,k)$ decreases in response to the increase of pruning percentage $p$, $E_{WG}$ decreases proportionally. The plot in the following link depicts the weight gradient energy $E_{WG}$ alongside the term in the R.H.S of (2), i.e. $C(\sigma_X,\sigma_W,p,k)$.
>
> https://drive.google.com/file/d/1tKiru-ZX9K_0Ocjeai74zdDLDqwjUGEc/view?usp=sharing
>
> As we have shown in the paper that $1/\sqrt{C(\sigma_X,\sigma_W,p,k)}$ follows an S-shaped pattern in response to $k$ (Remark 2), the observation that $E_{WG}$ relates proportionately to $C(\sigma_X,\sigma_W,p,k)$  validates the usefulness of SILO in this setting of 2-layer neural networks.

---

> > ### Author Response · Authors · 2023-06-20
> > **Author Responses to Reviewer d58D (2/2)**
> >
> > **(3) [Sensitivity of parameters in SILO]** We note that two key parameters in SILO are $q$ and $\beta$, where the former determines at which pruning cycle SILO enters the slow growth phase and the latter determines the shape of the S-curve. Please note that $\gamma$ is also computed from $q$ (see Section 4.3).
> >
> > In the main paper, to demonstrate the robustness of $q$ and $\beta$, we choose a fixed value of $q$ = 1 and  $\beta$ = 5 through all experiments (see Section 5.1) and demonstrate that this configuration outperforms existing benchmarks. To further verify the sensitivity of those two parameters, we conduct new experiments and demonstrate the sensitivity of these two parameters in the following link.
> >
> > https://drive.google.com/file/d/1T4ulRCKmeSWK3sUtHZqjBaDh05G8g1zX/view?usp=sharing
> >
> > The takeaway message from results shown in the link is three-fold: (i) The parameters of SILO can provide promising results (i.e., outperform benchmark schedules) within the suggested value range, i.e., 0 $\leq q \leq $ 3, 3 $ \leq \beta \leq$ 6 (see first 5 rows in Tables 8- 9 in the link). (ii) When the value of $q$ and $\beta$ fall outside the suggested range, SILO may fail to outperform benchmarks (see last 2 rows in Tables 8 - 9 in the link). (iii) It also suggests that $q$ and $\beta$ could be tuned within a relatively small range and does not require as much effort as tuning the hyperparameters.
> >
> > $\newline$
> >
> > Thanks again for your valuable comments. We hope our responses above address your concerns about the work, and all changes/results mentioned above will be updated in the revised version. Please feel free to add if you have any new doubts or comments.

---

### Review · Reviewer_zCPr · 2023-06-07

**Summary Of Contributions:**

This work proposes SILO, a learning rate schedule that follows a sigmoid type of curve, for better training of sparse networks. The authors motivate such a learning rate from a specific observation; as the network gets pruned the distribution of the gradients becomes more concentrated around small values, so the learning rate needs to appropriately increase. This is shown empirically in the case of iterative magnitude pruning of ReLU networks, where both the weight gradients as wells as the magnitude of the hidden representations (post ReLU) is reduced. The authors also provide theoretical arguments on why this is the case when considering Gaussian data and a two layer neural network with an infinite width hidden layer.  The main idea is that the gradient norm can be lower bounded by a function that depends on the learning rate and the activation norm, where the latter is also lower bounded by a function that depends on the sparsity ratio of the model. Therefore, if one desires a specific gradient magnitude, the learning rate needs to be adapted in a way that depends on the sparsity ratio and this relationship manifests as a sigmoid-type of curve. In practice, the authors define a learning rate schedule as $\delta \text(Sigmoid(\beta \log \frac{\gamma(p, q)}{1-\gamma(p, q)})) + \epsilon$, where $\gamma$ depends on the pruning ratio $p$ and $\delta, \beta, q, \epsilon$ are hyperparameters of the scheduler.

The authors show that this schedule does better than several learning rate schedulers used in practice in various tasks that involve pruning diverse set of models on CIFAR 10, 100 and Imagenet.


**Audience:**

Yes

**Broader Impact Concerns:**

No concerns.

**Claims And Evidence:**

No

**Requested Changes:**

The work is generally interesting and presents, in hindsight, an interesting observation. SILO ends up being a simple enough learning rate schedule that does provide some improvements upon the baselines. Having said that, the improvements seem marginal, especially given the relatively large standard deviation associated with each of the numbers; more often than not, the second best learning rate schedule has results that are within standard deviation of SILO.  Besides that, the proofs of the theoretical claims in the appendix could use a bit more polish as currently they can be hard to follow and some of the claims are unclear.

Based on these, the main requests I have / clarifications I would like on the experimental side are:
- Given the current status of the results and given that SILO does introduce more hyper parameters in the schedule, it would be interesting to see how robust SILO and the rest of the learning rate schedulers are with respect to their hyperparameters.
- Given that the main argument for SILO is that the gradient magnitudes change with pruning, it is peculiar that there are still benefits when doing Adam / RMSProp, which in general are invariant to the scale of the gradient. Perhaps one potential cause would be that the moment estimates might take a long time to adapt to the new gradient statistics that stem from pruning. In this case, it would be interesting to see what the improvements of SILO are when doing Adam / RMSProp and, e.g., the moments are “reset” on each pruning cycle.

And on the theoretical / claim side:
- The authors adopt non-standard definitions of “activation energy” and “weight gradient energy”, which are equivalent to squared L2 norms. I would suggest the authors to use the latter, more standard, definitions.
- At the end of page 17 I believe it should be $\mu_{trunc} = \int_{-\inf}^{\inf} x P_{trunc}(Z_j(X) = x)dx$
- At eq. 15 I believe that the first term should be  $\int_{-\inf}^{-\beta} W^2 P(W)dW$.
- The transition from eq. 10 to eq.11 is not clear
- At the proof of theorem 1 the authors use $Sign(W) = 0$ for $-\beta \leq W \leq \beta$ and $Sign(W) = 1$ otherwise. As the $Sign()$ is used primarily to get the actual sign of a variable, I would propose to instead use the indicator function, i.e., something like $I[|W| > \beta]$.
- At the extension of theorem 1 to the arbitrary depth it is mentioned below eq 30 that $\delta_1, \delta_2, …, \delta_M$ are independent of $X_1, W_1^{1k}, W_2^{k1}$, which is a peculiar statement as $\delta_1, …, \delta_M$ are functions of $X_1, W_1^{1k}$. Could the authors elaborate?
- At the proof of theorem 2 before eq. 46 the authors claim that $p(\alpha_l^k | h_i^k) = p(\alpha_l^k)$ which is once again peculiar as, from what I understand, $\alpha_l^k$ is the softmax probability for class $k$ which is a function of $W_2$ and the hidden layer $H^k$ (and $h_i^k$ is just one specific dimension of $H^k$).

And some various minor things:
- Some of the discussion points in the comparison at 5.2 do not align with the results at the table. For example, at 5.2 (3) the authors mention that SILO outperforms LR-Warmup by 1.9% but at Table 6 I see a 1.4% improvement (similarly for the 3.2% at $\lambda=5.72$, which at Table 6 is 2.3%).
- At (3) of page 12 it is mentioned that Tables 10-11 are in the Appendix whereas they are in the main text.



**Strengths And Weaknesses:**

Strengths
- The method and motivation intuitively make sense
- Mild improvements upon currently employed schedulers
- Results are architecture agnostic

Weaknesses
- The improvements seem to be marginal
- SILO introduces, relative to other common schedules, additional hyper parameters for tuning
- Some unclear claims in the proofs

---

> ### Author Response · Authors · 2023-06-20
> **Author Responses to Reviewer zCPr (1/3)**
>
> Dear Reviewer,
>
> Many thanks for your valuable time and effort in reviewing the paper. Special thanks for your constructive comments to the theoretical part. In the following, we clarify several points and present some new experimental results based on your suggestions. **We note that all clarifications and new results mentioned below will be updated in the revised version.**
>
> $\newline$
>
> **(1) [Robustness and sensitivity of parameters in SILO]** We note that two key parameters in SILO are $q$ and $\beta$, where the former determines at which pruning cycle SILO enters the slow growth phase and the latter determines the shape of the S-curve. Please note that $\gamma$ is also computed from $q$ (see Section 4.3).
>
> In the main paper, to demonstrate the robustness of  $q$ and $\beta$, we choose a fixed value of $q$ = 1 and  $\beta$ = 5 through all experiments (see Section 5.1) and demonstrate that this configuration outperforms existing benchmarks. To further verify the sensitivity of those two parameters, we conduct new experiments and demonstrate the sensitivity of these two parameters in the following link.
>
> https://drive.google.com/file/d/1T4ulRCKmeSWK3sUtHZqjBaDh05G8g1zX/view?usp=sharing
>
> The takeaway message from results shown in the link is three-fold: (i) The parameters of SILO can provide promising results (i.e., outperform benchmark schedules) within the suggested value range, i.e., 0 $\leq q \leq $ 3, 3 $ \leq \beta \leq$ 6 (see first 5 rows in Tables 8- 9 in the link). (ii) When the value of $q$ and $\beta$ fall outside the suggested range, SILO may fail to outperform benchmarks (see last 2 rows in Tables 8 - 9 in the link). (iii) It also suggests that $q$ and $\beta$ could be tuned within a relatively small range and does not require as much effort as tuning the hyperparameters.
>
> $\newline$
>
> **(2) [Experimental results using Adam and RMSProp]** We agree with your insightful comments and in fact, we have shown the performance of SILO using Adam and RMSProp in Section 6 (see part (3)). The results are provided in Tables 10 & 11 on page 12. In Table 10 (performance of using Adam), we observe that SILO increase the test accuracy from 75.2% (the benchmark performance using LR warmup) to 79.5% when  = 5.72. Similarly, when using RMSProp (see Table 11), the benchmark performance (i.e., using LR warmup) is increased from 67.3% to 72.5% when  = 5.72.

---

> > ### Author Response · Authors · 2023-06-20
> > **Author Responses to Reviewer zCPr (2/3)**
> >
> > **(3) [Theoretical clarifications]**
> >
> > (i): Yes, we are shifting to more conventional nomenclature for the average activation energy (activation L2-norm) and weight gradient energy (weight gradient L2-norm)
> >
> > (ii) and (iii): Yes, these are typos and have been corrected.
> >
> > (iv): We now provide the steps involved to yield (11) from (10). These steps have been added to the proof.  First, from (10), note that we can write $E_X[H_j(X)^2] = P(b_j\leq0)E_X[H_j(X)^2|b_j\leq0] + P(b_j>0)E_X[H_j(X)^2|b_j>0]$. Then, with $P(b_j\leq0)=P(b_j>0)=0.5$, $Z_j(X)\sim \mathcal{N}\left(b_j,\sigma_X^2\sum_{i=1}^d\left(W_1^{ij}\right)^2\right)$ (see just before (6)), and given the result in (10), we can write, $ E_X[H_j(X)^2]  = P(b_j\leq0)E_X[H_j(X)^2|b_j\leq0] + P(b_j>0)E_X[H_j(X)^2|b_j>0] \geq \frac{1}{2}\times\frac{1}{2}E_X[Z_j(X)^2] + 0 = \frac{1}{4}\left(b_j^2+\sigma_X^2\sum_{i=1}^d\left(W_1^{ij}\right)^2\right)$. We lastly have, $\frac{1}{N}\sum_{j=1}^{N}E_X[H_j(X)^2] \geq \frac{1}{4} \left (E_j[b_j^2]+\sigma_X^2\frac{1}{N}\sum_{j=1}^{N}\sum_{i=1}^d\left(W_1^{ij}\right)^2  \right)$, which is (11).
> >
> > (v) Sure, we have changed the writing to use the indicator function instead of the sign operator.
> >
> > (vi) What we meant was that $\delta_1,..\delta_M$ are each independent of $X_1$, $W_1^{1k}$ and $W_2^{k1}$ individually.  $\delta_1,..\delta_M$ are indeed functions of $X_1$ and $W_1^{1k}$, however, they need both $X_1$ and $W_1^{1k}$ to estimate. If we are only given $W_1^{1k}$, then we don’t get any additional information about $\delta_i$, as $X_1$ is unknown. Note that this is because $\delta_i$ is effectively related to the sign of the $i^{th}$ hidden neuron output at the first layer (before the non-linearity), and thus with only $X_1$ known and with only $W_1^{1k}$ known, its distribution is unchanged in each case. Thus, $\delta_1,..\delta_M$ are separately independent of $X_1$, $W_1^{1k}$ and $W_2^{k1}$. This argument has been clarified in the proof.
> >
> > (vii) We note that the weights of the last layer are a random variable as well, and the weight gradient energy estimated in Theorem 2 uses that fact (note the marginalization over a distribution of $W_2$ in (2)). As such, knowledge of $h^k_i$ will not yield any additional information about $\alpha^k_l$, because (a) the weights that map the entire $H^k$ to the output $\alpha^k_l$ are an independent Gaussian distribution as described in (2) and (b) $h^k_i$ is just one of infinite other hidden layer outputs each of which affect  $\alpha^k_l$ as well given a fixed $W_2$. Like before, we clarify that this independence is again between the individual $h^k_i$ and the softmax output $\alpha^k_l$. This argument has been clarified in the proof.

---

> > > ### Author Response · Authors · 2023-06-20
> > > **Author Responses to Reviewer zCPr (3/3)**
> > >
> > > **(4) [Performance improvement computation]** We note that the performance improvement is the relative improvement ((a - b)/b), but not the absolute improvement (a - b), where b is the starting quantity (e.g., LR warmup) and a is the ending quantity (e.g., SILO). We will highlight this part in the revised version.
> > >
> > > $\newline$
> > >
> > > **(5) [Location of Tables 11 - 12]** Yes, Table 11 and 12 are in the main paper, we are sorry about the misleading description (i.e., ‘Appendix’) of the table, we will update it in the revised version. Also, we will carefully look through the paper again to ensure the correctness of the whole paper.
> > >
> > > $\newline$
> > >
> > > Thanks again for your valuable comments. We hope our responses above address your concerns about the work, and all changes/results mentioned above will be updated in the revised version. Please feel free to add if you have any new doubts or comments.

---

### Review · Reviewer_ALQq · 2023-06-11

**Summary Of Contributions:**

In this paper, the focus lies on investigating the impact of learning rate (LR) schedules on network pruning. A novel LR schedule called SILO is introduced for network pruning, which exhibits enhanced generalization compared to existing LR schedule benchmarks across diverse network architectures.

**Audience:**

Yes

**Claims And Evidence:**

Yes

**Requested Changes:**

See in Strengths and Weaknesses.

**Strengths And Weaknesses:**

Strength:

In this study, the paper focuses on exploring the optimal selection of the learning rate during network pruning, supported by theoretical derivations. Additionally, a novel learning rate schedule named SILO is introduced for network pruning, showcasing superior generalization compared to existing benchmarks of learning rate schedules across a wide range of network architectures.


Weakness:

First, I find it difficult to comprehend the necessity of the fixed weight-gradient energy ($E_{WG}$). It would be helpful if the author could provide an explanation for its inclusion, along with relevant references to support its use.

Second, the assumptions made for Theorem 1 and 2 in this paper are excessively stringent. It assumes that (1) the weights follow a Gaussian distribution, (2) that the input to each layer also follows a Gaussian distribution, and (3) the neuron width is infinitely large *simultaneously*. To derive the S-shape trajectory using Theorem 1 & 2, this paper assumes that the weights consistently adhere to a Gaussian distribution throughout training, which is not practical. Furthermore, in deep neural networks, the input to the ($\ell$+1)-th layer can be viewed as the output of the ($\ell$)-th layer. However, with the ReLU activation function, assuming that the input of the ($\ell$)-th layer follows a Gaussian distribution does not guarantee that the input to the ($\ell$)-th layer will still follow a Gaussian distribution.

Third, it is crucial for this paper to establish a connection between its theoretical results and the improved generalization, which is the most significant numerical finding. Furthermore, providing a comparison with existing theoretical analyses of pruned networks would be essential in assessing the theoretical contributions of this paper.

Some additional questions:

Q1. Expanding the proposed methods to include structured pruning approaches would enhance computational efficiency when implementing them in real-world applications. It would be beneficial if the author could explore the extension of the methods to incorporate structured pruning techniques.

Q2. It would be highly valuable if the author could offer insights into the underlying intuition behind how adjusting the learning rate can result in improved generalization, considering it from an optimization perspective.

---

> ### Author Response · Authors · 2023-06-21
> **Author Response to Reviewer ALQq (1/2)**
>
>
> Dear Reviewer,
>
> Many thanks for your valuable time and effort in reviewing our manuscript. In the following, we outline changes to be made based on your suggestions. **We note that all clarifications and new results mentioned below will be updated in the revised version.**
>
> $\newline$
>
> **Weaknesses (1) & (3) and Q2**:
>
> We feel that Weakness (1) & (3) and Q2 are strongly related, and thus we answer them all with the detailed response below.
>
> **[Purpose of fixing weight gradient energy and relation to generalization performance]** Our initial motivation of the method is based on the series of observations in Section 3.1, where we find that average weight gradients undergo a significant reduction as the network gets pruned. Our intuitive hypothesis is that this reduction of weight gradients hampers the ability of the pruned network to fit the data well, i.e., it prevents the network from escaping local optimum points due to the smaller weight updates. The weight gradient energy term is specifically constructed to monitor the average change of network weights during a single weight update iteration from the gradient descent. To negate the impact of weight gradients decreasing during pruning, we enforce the constraint of fixed weight gradient energy to derive the learning rate trajectory.
>
> Furthermore, we note that the weight gradient energy is closely related to the gradient norm of a neural network, and there has been some recent studies (see [1]) which study the impact of gradient norm on generalization performance. It was found in [1] that the gradient norm effectively encodes a surrogate complexity of the network (Theorem 3.5), i.e., a smaller gradient norm hinders the ability of the network to fit the data. In our case, however, there are two things simultaneously happening when we prune networks:
>
> (1) The pruned networks themselves lose a significant amount of their base complexity, affecting their ability to fit labels successfully.
>
> (2) The weight gradients undergo a significant reduction on average (see Figure 1 in section 3.1), negatively affecting the gradient norm even more, and reducing the surrogate complexity of the pruned network even further. This is intuitively expected, as a smaller weight gradient norm implies that a smaller part of the classifier’s function space is reachable during the training process, which reduces the effective complexity of the classifier.
>
> Thus, as the network is pruned, we should expect the effective complexity of the network to decrease quite significantly due to the two reasons mentioned above, which is expected to negatively impact the training accuracy of the network. Our hypothesis is that this then directly impacts the test accuracy as well, as a network that underfits will eventually yield a lower test accuracy and thus poor generalization performance. To avoid this situation, we impose a constraint of fixing the weight gradient energy by increasing the learning rate as the network is iteratively pruned, so that it increases the surrogate complexity of the pruned network, which avoids underfitting and improves training accuracy. Note that Theorem 1 in [4] and Theorem 5.2 in [3] also showcase how increasing learning rate can increase the surrogate complexity of a network.
>
> $\newline$
>
> **[New experiments]** To verify the hypothesis that fixing weight gradient energy helps the network to train better, we conduct new experiments to examine the training accuracy of pruned networks. Please see plots in the following link.
>
> https://drive.google.com/file/d/12USfpSAoADp_U1y1rj074mQwTcB-LlBL/view?usp=sharing
>
> In the figure above, we examine the training accuracy of ResNet-20 and VGG-19 during iterative pruning using global magnitude on CIFAR-10. We find that SILO helps to obtain a better training accuracy, which automatically improves test accuracy as well. These results directly reaffirm our hypothesis that without learning rate regulation, the training accuracy of pruned networks undergo a significant reduction, impacting its overall generalization performance as well.
>
> $\newline$
>
> **[Other pruning theory]** Our objective with the theoretical results was to mainly show that an S-shaped learning rate curve emerges when attempting to ensure that the weight update magnitudes stay the same in response to network pruning, validating our original empirical observations and motivation. While we have seen many theoretical analyses of pruning networks in general, we did not find any other theoretical works that specifically derive a learning rate pattern by fixing the weight-gradient energy quantity that we analyze in this work. However, as mentioned previously, we do find that there are many works that show how the gradient norm has an effect on the overall complexity and generalizability of neural networks, and we are adding these references and perspectives to the paper.

---

> > ### Author Response · Authors · 2023-06-21
> > **Author Response to Reviewer ALQq (2/2)**
> >
> > **Weakness (2):**
> >
> > **[Strictness of Assumptions]** We have now relaxed the assumption of infinite hidden nodes, yielding an extended version of Theorem 1 which yields a probabilistic lower bound on the average activation energy. We now consider a finite number of hidden neurons $N$, in the hidden layer $H$, thus we have that $N < \infty$. Furthermore, given initialized network weights and biases of the first layer in $W_1$, we let $w_m$ denote the maximum valued weight/bias. Then, after $k$ iterations of pruning, it holds that
> > $Pr\left(4E_{AA}(H) > C(\sigma_X,\sigma_W,p,k)-\epsilon \right) \geq 1 - e^{- \left(2N\epsilon^2/w_{m}^4(1+d\sigma_x^2)^2\right)}$
> >
> > where $C(\sigma_X,\sigma_W,p,k)$ has the same definition as in Theorem 2 (which is the R.H.S of the original Theorem 1). Thus, the above is a probabilistic lower bound on the average activation energy when the number of hidden nodes $N<\infty$. Note that this result does not change the takeaway of our Theorems, as we still analyze the quantity $C(\sigma_X,\sigma_W,p,k)$, which is still the only quantity that changes in response to network pruning percentage. We have added this result and its proof in the Supplementary materials (see Proposition 2).
> >
> > We further note that we have empirically tested the results of Theorem 1 and 2, using 2-layer neural networks with finite hidden nodes (1000) and 100 input nodes, using a fixed $\sigma_x$ and $\sigma_w$. For Theorem 1, we record the average activation energy against the percentage of pruned nodes in the remaining network, and find that in each case, the L.H.S of (1) is smaller than the R.H.S. The plot is provided in the following link.
> >
> > https://drive.google.com/file/d/1PC9bWpozD1BFyy0rw3QP45_qVusEMlX6/view?usp=sharing
> >
> > For Theorem 2, we find that the average weight gradient energy (L.H.S of (2)) very closely follows the trend predicted by the R.H.S of (2) for a fixed choice of the learning rate $\alpha$. The plot is shown in the following link.
> >
> > https://drive.google.com/file/d/1tKiru-ZX9K_0Ocjeai74zdDLDqwjUGEc/view?usp=sharing
> >
> > Note that to obtain a consistent trend, the average weight gradient energy was computed over 500 separate runs of each individual weight update iteration. Please note the scale differences between the two quantities.
> >
> >
> > $\newline$
> >
> > **[Gaussian Assumption for Multi-layered networks]** Yes, you are right in suggesting that the Gaussian assumption will not hold when adding more layers, as the distribution of the $l+1^{th}$ layer need not be a Gaussian, given the distribution of $l^{th}$ layer is a Gaussian. To circumvent this issue, we had worked around the concept of effective weight, with which we were able to prove a similar result as in Theorem 1 for fully connected networks of arbitrary depth. In our submitted version, we had showcased the result for multi-layered networks of arbitrary depth in the Supplementary Materials in Corollary 1 (renamed to Proposition 1 in revised paper).
> >
> > $\newline$
> > $\newline$
> >
> > **Q1:**
> >
> > **[Extension to Structured Pruning]**. In Section 3.2, we have examined the distribution of weight gradients using structured pruning and shown that structured pruning suffers from the issue of decreasing weight gradients as well (see Fig. 6 in the Appendix for more details). This suggests that SILO could be applicable to structured pruning as well.
> >
> > We conduct new experiments and compare the performance of SILO to LR-warmup using L1 norm filter pruning [2] on ResNet-20 and VGG-19 using CIFAR-10. The results are summarized in the following link
> >
> > https://drive.google.com/file/d/1n9vqBU6L3cVRK9lvuFs9Q8m2DQ2mh0L7/view?usp=sharing
> >
> >
> >
> > We observe that SILO still outperforms the benchmark LR schedule, leading to an improvement of 2.0\% (compare SILO at row (ii) to LR-warmup at row (i) when $\lambda$ = 7.29). We note that the performance improvement is not as significant as that of working with unstructured pruning. We suspect it is due to that the distribution of weight gradients may change differently when the network is structurally pruned. In such a case, SILO may need to be revised for a much larger performance gain.
> >
> >
> > [1] Gat, Itai, et al. "On the Importance of Gradient Norm in PAC-Bayesian Bounds." Neurips 2022
> >
> > [2] Hao Li, et al. Pruning filters for efficient convnets. ICLR, 2017
> >
> > [3] Luo, Xuanyuan, Bei Luo, and Jian Li. "Generalization Bounds for Gradient Methods via Discrete and Continuous Prior."  Neurips 2022
> >
> > [4] Wang, Ziqiao, and Yongyi Mao. "On the Generalization of Neural Networks Trained with SGD: Information-Theoretical Bounds and Implications.", ICLR 2022
> >
> > $\newline$
> >
> > Thanks again for your valuable comments. We hope our responses above address your concerns about the work, and all changes/results mentioned above will be updated in the revised version. Please feel free to add if you have any new doubts or comments.

---

### Author Response · Authors · 2023-06-23
**New Revision Uploaded based on Reviewers' Suggestions**

Dear Action Editor and Reviewers,

On behalf of all the contributing authors, I would like to express our sincere appreciations of your professional work and reviewers' constructive and insightful comments. Based on your suggestions, we have revised the manuscript extensively and added new experimental results to better convey the ideas of the whole paper. We have uploaded a new version of our manuscript and summarize the changes as follows:

**[New experimental results]**

1. Reviewer d58D & Reviewer zCPr: We have added new experiments to demonstrate sensitivity and robustness of parameters in SILO (see Section 5.3 in the main paper).
2. Reviewer d58D: We have added new experiments to examine the effect of SILO on weight update (see Section C2 in the Appendix).
3. Reviewer ALQq: We have added new performance evaluation on AP using structured pruning (see Section 6, point 4 in the main paper).

 **[Theory related changes and additions]**

1. Reviewer D58d: We have added the experiments which verify the findings in Theorems 1 and 2 of the main paper in Section B of Appendix. More experimental details have been included in Section B.
2. Reviewer zCPr: We have made all requested corrections to the typos in the proofs, including adding the necessary transitions steps from (10) to (11), and further explaining the reasoning behind the independence of the random variables involved in the proofs of Theorem 2 and Corollary 1 (which is now called Proposition 1 in the updated paper).
3. Reviewer zCPr: We also have replaced the notions of average activation energy and weight gradient energy by “average activation norm” and “weight gradient norm”.
4. Reviewer ALQq: We have added a section describing how intuitively SILO will be expected to improve generalization performance (see Section C.3), containing references to support the hypothesis that SILO would improve the surrogate complexity of the network and thus improve training performance, which is also verified via additional experiments (Figure 10).
5. Reviewer ALQq: We also have added more reasons behind our fixed weight-gradient energy assumption (see Section C.3), based on our response to the reviewer, and we have also clarified this assumption in the updated Remark 2 of the main paper.
6. Reviewer ALQq: We also have added Proposition 2, which relaxes the assumption of infinite hidden nodes in Theorem 1 and outputs a probabilistic bound.

We hope these changes can address reviewers' concerns. Please feel free to add if you have any new comments or doubts. Thanks again for your valuable time.

---

### Decision · Action_Editors · 2023-07-19

**Recommendation:** Accept as is

**Comment:**

This paper studies the learning rate schedule for iterative neural network pruning. Motivated by empirical observation and theoretical motivation, it proposes an S-shaped learning rate schedule during pruning iterations. The approach shows performance gains across various datasets, architectures, and optimizers.

Three reviewers provided detailed and constructive comments for this paper and the authors responded with additional experimental results and theoretical analysis. Most issues, except one remaining assumption under the theoretical results, were well addressed in the revised version. Nevertheless, the authors have provided empirical evidence in support of the theoretical motivation (e.g. Fig. 3) as well as the effectiveness of the LR schedule (e.g. Table 11). These contributions are meaningful for the community and AE suggests the Accept decision.

**Audience:**

Yes. Researchers working on neural network pruning would have interests in this paper.

**Claims And Evidence:**

Yes. This paper provides extensive experiments, validating the theoretical motivations, superior performance, and hyperparameter robustness of the proposed method.